# OpenReview forum: "TOM-SWE: User Mental Modeling For Software Engineering Agents"
_ICLR.cc/2026/Conference — Submitted to ICLR 2026_

### Official Review · Reviewer_cL3T · 2025-10-28

**Soundness:** 2
**Presentation:** 3
**Contribution:** 3
**Rating:** 4
**Confidence:** 4

**Summary:**

The paper introduces a framework for incorporating theory-of-mind modeling into a software engineering agent (ToM-SWE). The approach involves delegating to a special theory-of-mind agent for intent modeling and storing user preference memory. The framework is evaluated on ambiguous SWE-bench and a newly introduced benchmark, stateful SWE-bench, as well as a live user study. The stateful SWE-benchmark is created by mining user profiles from real user interactions on the OpenHands platform and creating a user simulator.

**Strengths:**

- The paper makes a good case for improving user modeling for interactive agents software engineering scenarios.
- The paper establishes simple but meaningful categories for common user preferences in software engineering agents.
- The proposed augmentation with the ToM agent performs well on interactive SWE-bench variants, as well as receiving good feedback in an online trial with human software engineering.

**Weaknesses:**

- The paper claims that the advantage of the two agent solution is ”reduced context distraction and specialized optimization”. This seems plausible but I believe the paper provides limited evidence beyond the simple RAG baseline (which already performs quite well with Claude 4). For example, I’d imagine one could prompt the main agent better to think about user intents, or the user profile analysis could be an offline step (aggregating past sessions +  user profile → updated user profile), that is included in the context of the main coding agent.
- The results of the online user study is problematic without a baseline or control group that uses the regular CodeAct product. Specifically, metrics like accept/partial/reject rates are hard to interpret without knowledge of the baseline.

**Questions:**

- Do the two agents ever communicate more than once for a single user query? I.e. how often does the agent call the `consult_tom` function in response to a user query? And does the response differ when called multiple times for the same user query?
- It would be really helpful to see a complete timeline of the back-and-forth between the main agent and the ToM agent across multiple user sessions.
- Can I interpret ambiguous SWE-bench to be in some sense a special case of stateful SWE-bench but with a fixed and particular “multi-step” type of user profile?

---

> ### Author Response · Authors · 2025-11-24
>
> We thank you for your review. We are encouraged that you found our paper makes a good case for user modeling. We have addressed each of your weaknesses and questions below. We have also updated the draft accordingly (see the blue updated text).
>
> > The paper claims that the advantage of the two agent solution is ”reduced context distraction and specialized optimization”. This seems plausible but I believe the paper provides limited evidence beyond the simple RAG baseline (which already performs quite well with Claude 4). For example, I’d imagine one could prompt the main agent better to think about user intents, or the user profile analysis could be an offline step (aggregating past sessions + user profile → updated user profile), that is included in the context of the main coding agent.
>
> We would like to clarify that we already explicitly prompt both RagCodeAct and CodeAct to interact with and reason about the user (line 257-258). Our prompt for the agents states:
>  “*You are encouraged to use non-tool_call actions to engage with the user/me… including providing progress updates, asking for clarification, etc. Before issuing the finish action, you may communicate with the user/me to resolve the issue better.*”
> We also observed both agents engaging with the simulated user (10% for the CodeAct agent and 18% for the RagCodeAct agent). We also include the full prompts for different agents in Appendix A11.
>
> We agree that an explicit instruction to continuously maintain such a file is an additional baseline, and we conducted a preliminary experiment (N=100) using Claude 4 Sonnet (new experiment). We equipped the single SWE-agent with the full set of ToM prompts and loaded the `Openhands.md` file, containing summarized user preferences, directly into the agent's context window.
>
> Results: This single-agent setup achieved a 27.17% resolution rate, which is notably lower than the 38.4% achieved by our simpler RAGCodeAct baseline. Our hypothesis is that requiring a single agent to simultaneously manage complex coding tasks and explicit user profile maintenance introduces "context overload," degrading its ability to reason effectively about the code. Given the clear signal from this 100-sample run and the high computational cost of a full-scale baseline evaluation (~$1,500 per run), we believe this sufficiently demonstrates that a multi-agent architecture has its unique advantages.
>
> > The results of the online user study is problematic without a baseline or control group that uses the regular CodeAct product. Specifically, metrics like accept/partial/reject rates are hard to interpret without knowledge of the baseline.
>
> We respectfully disagree with the premise of this weakness; the human study is not intended to measure downstream code-acceptance rates; its goal is to evaluate whether the ToM agent’s suggestions are useful. The most direct way to measure this is to ask users about the perceived usefulness of each suggestion. Thus, accept/partial/reject responses are appropriate for the phenomenon we seek to evaluate. Comparing against typical code/commit acceptance rates is a great idea for future directions.
>
> ### Questions
>
> > Do the two agents ever communicate more than once for a single user query? I.e. how often does the agent call the consult_tom function in response to a user query? And does the response differ when called multiple times for the same user query?
>
> Yes, the two agents communicate more than once for a single user query (that happen on average 14% of the time). Within a session, the SWE agent may call consult_tom multiple times, even without additional user input, because it can independently seek clarification from the ToM agent. The response differs when called multiple times for the same user query.
>
> > It would be really helpful to see a complete timeline of the back-and-forth between the main agent and the ToM agent across multiple user sessions.
>
> We agree. We updated the paper with a complete timeline of the back-and-forth (Figure 7). It shows the process of how the ToM agent “learned” the user’s preferences in the previous session could help figure out the user's intent better in the following session.
>
> > Can I interpret ambiguous SWE-bench to be in some sense a special case of stateful SWE-bench but with a fixed and particular “multi-step” type of user profile?
>
> Partially, but not exactly. As shown in Figure 3, the original SWE-bench issue is converted into a vague user instruction (we updated the draft to clarify this in Figure 3). As noted in the main text (lines 223–235):
> “Differing from Ambiguous SWE-bench, the user simulator in Stateful SWE-bench conditions on the unique user profile… Meanwhile, agents have access to previous conversation histories with the same user profile…”
> Thus, the ambiguous SWE-bench resembles a simplified case, but lacks the personalized and stateful components central to Stateful SWE-bench.

---

### Official Review · Reviewer_ofCv · 2025-10-31

**Soundness:** 2
**Presentation:** 3
**Contribution:** 2
**Rating:** 4
**Confidence:** 3

**Summary:**

1. Multi (paired) agent SWE-harness, one managing user goals/memory/preferences from interaction history, with the other agent managing actual task performance but taking in instruction/context from ToM agent.

1a. In practice, this looks like extra verification and receiving extra guidance from a supervising agent.

2. Contribution of Stateful SWE benchmark, which incorporates an LLM powered user simulator on top of SWE-bench style issues, where agents are asked to resolve the issues while also measuring user satisfaction.

2a. The authors collect sessions to generate 15 “user” profiles and use GPT-5 to transform the SWE-bench issue statement into a user instruction aligned with a given user profile. This user instruction is used as the new initial issue statement into SWE-bench style evaluation, and the SWE-agent is allowed interaction with the simulated user as well (Line 129).
2b. In addition to resolution rate, this benchmark also measures user rating and satisfaction (Figure 3)

3. Evaluation on stateful SWE benchmark and ambiguous SWE-bench using the OpenHands platform.

**Strengths:**

1. Novel hierarchical agentic architecture that helps improve user satisfaction on SWE-style tasks.
2. Contribution of a new benchmark, Stateful SWE benchmark, which evaluates how well agents sustain meaningful interactions over time (evaluates long-term memory demands) with an interesting user-simulator based approach.
3. Human study with real-world developers to validate ToM agent’s importance is very novel and shows strong results of ToM and user satisfaction.

**Weaknesses:**

1. From my read of Section 3, I could not immediately tell what the difference of stateful SWE-bench from SWE-bench was in terms of problem_statement or other task inputs and outputs: I think this section would benefit from a diagram showing how SWE-bench issues were mapped and a specific example of what an instance of the mapping looks like (i.e. is the problem_statement modified and if so, in what way?).

1a. Also, how big is this new benchmark, is it 15 x 500 = 7500 total instances? If that’s the case, it seems like if a base model can already solve an instance in SWE-bench, there may be some bias or non-representative performance in the modified SWE-bench (may not be the case, but more clarity in the main paper here would help given it’s a core contribution).

2. I understand why one would use ambiguous/stateful SWE-bench type benchmarks to evaluate user intent (as normal SWE-bench does not really evaluate interactive user interaction or long-term memory). However, the paper presents a very core contribution of a huge increase of ToM-CodeAct over CodeAct, which doesn’t seem that surprising for the following:

2a. My mental model is that, seems that most of CodeAct (and likely CodeRAGAct) underperformance here is strictly from the prompt being somehow underspecified in this new benchmark and CodeAct can’t interact with the user for clarification or retrieve previous trajectories?

2b.Could the authors elaborate more on this point: specifically (1) if a simulated user is part of the benchmark and if at evaluation time, the agent can interact with this and (2) whether CodeAct tries to interact with the simulated user? If not, this would attribute the under-performance to ambiguity in the initial problem statement, rather than ToM’s architecture.

3 The paper lacks deeper analysis into the tradeoffs between user satisfaction and task resolution.

3a. In the paper you mentioned that “user satisfaction does not equal solving the task”: this seems like there are two modes: (1) “we solve the task but in a way that makes the user unhappy” and (2) “we don’t solve the task but the user is happy”? (1) seems more harmless, but (2) seems considerably more risky: is higher really better here since the user might be fooled into thinking the task is solved?

3b. Is there correlation analysis between resolution rate and user satisfaction to clarify this more?

**Questions:**

1. Diversity of 15 developer profiles: is there selection bias here from actual programmers, or analysis to observe how diverse these are in practice?
2. Nit: Figure 5 is hard to read and feels like it would be easier to read if inverted in color (darker/more intense color is higher number)
3. Are there numbers comparing CodeAct to ToM acceptance numbers (or ToM to any baseline accept numbers)? The paper seems to primarily only present accept or reject, would be good to contextualize this against normal accept/reject numbers.

---

> ### Author Response · Authors · 2025-11-24
>
> We thank you for your review of our paper. We have addressed each of your weaknesses and questions below. We also updated the draft accordingly (see the blue updated text).
> > From my read of Section 3, ...I think this section would benefit from a diagram showing how SWE-bench issues were mapped and a specific example of what an instance of the mapping looks like (i.e. is the problem_statement modified and if so, in what way?).
>
> As shown in Figure 3, the original SWE-bench issue is transformed into a vague user instruction (we have updated the Figure 3 caption to make things clear). As mentioned in our main text (line 223-235), the user simulator in Stateful SWE-bench is conditioning on the unique user profile. Meanwhile, agents have access to corresponding previous conversation histories with the same user profile.
>
> > 1a. Also, how big is this new benchmark, is it 15 x 500 = 7500 total instances? ... may not be the case, but more clarity in the main paper here would help given it’s a core contribution).
>
> Thanks for pointing this out, we agree and took this into account in our design. Specifically, to account for the individual complexity of the SWE-bench instances, we only include each instance once in our new benchmark; we randomly pair each instance with a profile sampled from one of our 15 profiles. Thus, the size is 500, just like the original SWE-bench.
>
> >  I understand why one would use ambiguous/stateful SWE-bench type benchmarks to evaluate user intent (as normal SWE-bench does not really evaluate interactive user interaction or long-term memory). However, the paper presents a very core contribution of a huge increase of ToM-CodeAct over CodeAct, which doesn’t seem that surprising for the following: 2a. My mental model is that, seems that most of CodeAct (and likely CodeRAGAct) underperformance here is strictly from the prompt being somehow underspecified in this new benchmark and CodeAct can’t interact with the user for clarification or retrieve previous trajectories? ...
>
> We believe there is a misunderstanding. During evaluation, we **not only allow** both the CodeAct and RagCodeAct agents to interact with the simulated user, but we **explicitly encourage** such interactions through an additional prompt: “*You are encouraged to use non-tool-calls actions to engage with the user/me, including providing progress reports, answering questions, asking for clarification, etc. Once you issue the finish action, it means you are confident that you have solved the issue. Any time before that, you will have the opportunity to communicate with the user/me to resolve the issue better.*” In practice, we observed that both CodeAct and RagCodeAct interacted with the user during evaluation. In addition, RagCodeAct can retrieve previous trajectories, further supporting its ability to engage effectively. (Please check the updated draft on Appendix A.7.5 for more detailed prompts.)
>
> > The paper lacks deeper analysis into the tradeoffs between user satisfaction and task resolution. 3a. In the paper you mentioned that “user satisfaction does not equal solving the task”: this seems like there are two modes: (1) “we solve the task but in a way that makes the user unhappy” and (2) “we don’t solve the task but the user is happy”? (1) seems more harmless, but (2) seems considerably more risky: is higher really better here since the user might be fooled into thinking the task is solved? 3b. Is there correlation analysis between resolution rate and user satisfaction to clarify this more?
>
> This is an important point, and we have examined it carefully. For the 44 cases where the Claude 4 ToMCodeAct agent received high user satisfaction (H) despite failing the task (F), our manual analysis shows that users were rewarding the quality of the process, not being misled about the outcome. Specifically, users valued:
> * Systematic, well-targeted investigation with clear technical reasoning (61% of F+H cases)
> * Clear communication and visible progress toward a solution (39% of F+H cases)
> In other words, this reflects a “good-effort bonus” where users appreciate a strong technical approach and good communication, even if the final fix is not achieved.
> Our quantitative analysis further shows that user satisfaction is strongly aligned with task success:
> Statistical evidence:
> * Pearson r = 0.74 (p < 0.001), indicating a very strong correlation
> * R² = 0.55, meaning task resolution explains 55% of the variance in satisfaction
>
> These results show that although users sometimes reward strong problem-solving processes, task success remains the dominant driver of satisfaction (we have updated our draft to reflect this; lines 351-362).

---

> ### Author Response · Authors · 2025-11-24
>
> ### Questions:
>
> > Diversity of 15 developer profiles: is there selection bias here from actual programmers, or analysis to observe how diverse these are in practice?
>
> The selection is grounded in actual programmer behavior, derived from internally collected real-world programmer sessions interacting with coding agents (we include some examples of developer profiles in Appendix A.10).
>
> While our profiles are grounded in real-world programmer sessions, capturing the full spectrum of developer behavior is inherently difficult due to the scarcity of large-scale, fine-grained coding interaction datasets. We view our current set as a strong initial baseline and believe that expanding this diversity to address potential bias is an important direction for future work.
>
> > Nit: Figure 5 is hard to read and feels like it would be easier to read if inverted in color (darker/more intense color is higher number)
>
> We have updated the figure in the draft.
>
> > Are there numbers comparing CodeAct to ToM acceptance numbers (or ToM to any baseline accept numbers)? The paper seems to primarily only present accept or reject, would be good to contextualize this against normal accept/reject numbers.
>
> Our human study here targets the usefulness of the ToM agent’s suggestions rather than downstream code acceptance. Because the study was fully in-the-wild, users chose their own tasks and integrated code at their discretion. Due to privacy concerns, we would only be able to determine whether users accept/modify/reject the ToM agent’s suggestions. Therefore, we could not reliably measure the final acceptance rates of generated edits and compare different agentic setups. We updated this in our paper (lines 398-400).
>
> Please let us know if you have any additional questions. If you think we have sufficiently addressed your points, we kindly ask that you consider updating your score.

---

### Official Review · Reviewer_XBeR · 2025-11-01

**Soundness:** 2
**Presentation:** 1
**Contribution:** 2
**Rating:** 4
**Confidence:** 4

**Summary:**

The paper provides:

- A multi-agent setup that includes a SWE agent paired with a ToM (theory of mind) agent, where the latter attempts to help to resolve ambiguities and improve the adherence of the SWE-agent to coding styles and preferences of the user.
- A new benchmark "stateful SWE-bench"
- A human study that tests the ToM-SWE agent with real human developers

**Strengths:**

The issue that is addressed by the contribution is relevant: Human work and interaction with agents is sequential, and having agents learn and persist user preferences across individual tasks is an important part.

To measure performance on sequential tasks, the paper introduce a new benchmark SWE-bench sequential, that allows to study tasks in sequence, and to model interaction with an opinionated human developers.
While some details are still unclear to me, this seems like an original and useful contribution.

**Weaknesses:**

## Readability

* For some time I did not understand what the actual score from Stateful SWE-bench score as shown in Fig. 4 actually is. The paper states (Section 3) it's the "user simulator satisfaction scores" and points to appendix A.6.2, but A.6.2 lists 5 different scores. But then in Fig. 4., what is shown (if I follow the main text) is actually the unittest based resolution rate and the "satistfaction score" is an extra score that is shown in Table 1. This could be make clear.
* Fig 1: I really like the screenshots, but I still don't understand what's happening above with the arrows (why is the normal SWE-agent setup connected by an arrow to the ToM-SWE setup? Aren't they two separate examples?)
* Fig 5: Had a hard time to understand what the numbers mean here. Maybe there's also a few simpler things you could have considered, for example by just comparing the averages of user satisfaction for the case of resolved vs unresolved task?
* Fig 6: Very hard to understand (especially without reading the main text in detail): I assume x axis is avg. ToM cost persession as opposed to the total cost per session? Is the dashed orange line a linear regression? I also think the plot might be easier to understand when not color coding model providers (the information is already in the labels) and potentially removing the bar chart inset. Would also be good to mention which resolved score this is in the plot fig description.

**Questions:**

* How does the performance of the ToM SWE-agent compare to just having a CLAUDE.md file with the user profile that is attached to the conversation history? Perhaps even with a setup where the agent is told it can also edit the same file to incorporate user feedback (and is told to do so at the end of each trajectory). I wonder if a simpler single-agent memory system without a multi-agent setup can lead to similar success.
* In the Ambiguous SWE-bench paper, there was a helpful plot showing resolve rates for their benchmark for three settings hidden (underspecified), interaction (agent gets to reqeust information), and full (fully specified). A similar plot might be helpful for the sequential benchmark as well.
* The appendix is very loose on details about the stateful SWE-bench benchmark. Without showing the prompt templates it's not possible to really understand what the user simulator satisfaction score is. Ideally, I would like to also see some concrete example outputs of the LM judge.
* It would also be very helpful to include a complete example of a developer profile instead of highly abbreviated summaries.
* Section 5: Do I understand it correctly that the user triggers the TOM agent explicitly with `/tom_give_suggestions` but the SWE agent also can do it? For the data shown in Fig. 7 how many times did the SWE agent ask and how often did the user explicitly request the tom agent? (and I assume even if the SWE agent requests that information itself, the user is still prompted to accept/modify/reject?)
* I did not understand the categories (Code understanding, development, etc.). Is this the overall task the TOM-SWE agent is given? Or is this specific to the interaction between the ToM and the user.
* It seems like the acceptance rate of ToM-SWE agent consultations only captures whether or not these conversation replies were deemed correct by the users, but not if they had a significant impact on the acceptance of edits that were finally made by the SWE agent. Or in other words, do you have any metrics from the human study that show the overall satisfaction rate of TOM-SWE agent compared with the normal CodeAct satisfaction rate?

---

> ### Author Response · Authors · 2025-11-24
>
> Thank you for the review and detailed suggestions. We have addressed your concerns about readability and clarity in our updated draft (see the blue updated text). We will first focus on answering the questions, and we will further explain each point of the concerns in readability to provide more context.
>
> ### Questions:
> > How does the performance of the ToM SWE-agent compare to just having a CLAUDE.md file with the user profile that is attached to the conversation history? Perhaps even with a setup where the agent is told it can also edit the same file to incorporate user feedback (and is told to do so at the end of each trajectory). I wonder if a simpler single-agent memory system without a multi-agent setup can lead to similar success.
>
> Our RAGCodeAct baseline already equips the agent with access to prior user conversations and the ability to summarize user profiles into a local Openhands.md file.
>
> We agree that an explicit instruction to continuously maintain such a file is an additional baseline, and we conducted a preliminary experiment (N=100) using Claude 4 Sonnet (new experiment). We equipped the single SWE-agent with the full set of ToM prompts and loaded the Openhands.md file, containing summarized user preferences, directly into the agent's context window.
>
> Results: This single-agent setup achieved a 27.17% resolution rate, which is notably lower than the 38.4% achieved by our simpler RAGCodeAct baseline. Our hypothesis is that requiring a single agent to simultaneously manage complex coding tasks and explicit user profile maintenance introduces "context overload," degrading its ability to reason effectively about the code. Given the clear signal from this 100-sample run and the high computational cost of a full-scale baseline evaluation (~$1,500 per run), we believe this sufficiently demonstrates that a multi-agent architecture has its unique advantages.
>
>
> > In the Ambiguous SWE-bench paper, there was a helpful plot showing resolve rates for their benchmark for three settings hidden (underspecified), interaction (agent gets to reqeust information), and full (fully specified). A similar plot might be helpful for the sequential benchmark as well.
>
> Our Fig. 4 includes both the full and interaction settings, since the hidden is only aiming at revealing the importance of interaction given underspecified instructions. From the Ambiguous SWE-bench, we know interaction already improves over hidden, so here our focus is on how different agents support better interaction with users under underspecified instructions.
>
>
> > The appendix is very loose on details about the stateful SWE-bench benchmark. Without showing the prompt templates, it's not possible to really understand what the user simulator satisfaction score is. Ideally, I would like to also see some concrete example outputs of the LM judge.
>
> We have revised the draft, and the prompt for the user simulator is available in the appendix section A.9 with a concrete example of the user judgement. We also added a concrete example of the simulator user profile in the appendix section A.10 that will be used as a system prompt for the LLM powering the user simulator.
>
> > Section 5: Do I understand it correctly that the user triggers the TOM agent explicitly with /tom_give_suggestions but the SWE agent also can do it? For the data shown in Fig. 7 how many times did the SWE agent ask and how often did the user explicitly request the tom agent? (and I assume even if the SWE agent requests that information itself, the user is still prompted to accept/modify/reject?)
>
> Both users and the SWE-agent may call the ToM agent. In our user study, only 9.7% of the time the users invoked it manually. For evaluation, users were prompted to accept/modify/reject the ToM suggestions so we could measure perceived usefulness.
>
> > I did not understand the categories (Code understanding, development, etc.). Is this the overall task the TOM-SWE agent is given? Or is this specific to the interaction between the ToM and the user.
>
>
> The categories are summarized from the collected 209 coding sessions from the user study with the help of LLM (the summarization prompt is added to the revised draft in the appendix section A. 11). We did not give users any predefined tasks; the 17 professional developers chose their own tasks, aligning with their daily work.

---

> ### Author Response · Authors · 2025-11-24
>
> ### Readability Issues
>
> > For some time I did not understand what the actual score from Stateful SWE-bench score as shown in Fig. 4 actually is. The paper states (Section 3)...
>
> Your interpretation is correct: the scores in Fig. 4 correspond to the standard SWE-bench instance evaluation (i.e., unittest-based resolution rate). The user-simulator satisfaction scores appear in Table 1. We have updated the draft to make that clear (line 269)
>
> > Fig 1: I really like the screenshots, but I still don't understand what's happening above with the arrows (why is the normal SWE-agent setup connected by an arrow to the ToM-SWE setup? Aren't they two separate examples?)
>
> They are two separate examples. The arrow is intended to illustrate how different agents would respond to the same underspecified instruction (we have updated Fig. 1's caption to highlight that).
>
> > Fig 5: Had a hard time to understand what the numbers mean here. Maybe there's also a few simpler things you could have considered, for example by just comparing the averages of user satisfaction for the case of resolved vs unresolved task?
>
> We are mainly aiming at showing there’s a difference between making the user satisfied and resolving the tasks (i.e., passing the unit tests). We have updated the caption of Fig 5 to make the trend easier to understand. Directly averaging the numbers risks smoothing the important trend in the data.
>
> > Fig 6: Very hard to understand (especially without reading the main text in detail): I assume x axis is avg. ToM cost per session as opposed to the total cost per session? Is the dashed orange line a linear regression? ...
>
> Yes, for Figure 6, the x-axis is avg. ToM cost per session; the resolved scores are the standard SWE resolution rate. The orange dashed line is a visual baseline for the cost–performance tradeoff. By using the slope of the delta of resolved rates divided by the delta of the cost across all the models we evaluated, it gives a quick yardstick: points above the dashed line are more cost‑effective than this baseline trend; points below are less. It’s not a regression fit. We have updated Fig 6's caption to make things clear.
>
> Please let us know if you have any additional questions. If you think we have sufficiently addressed your points, we kindly ask that you consider updating your score.

---

### Meta-Review · Area_Chair_oUCv · 2026-01-05

**Summary:**

This paper introduces ToM-SWE, a dual-agent framework that enhances software engineering agents with theory-of-mind capabilities to better infer and adapt to user intent. By pairing a primary SWE agent with a lightweight ToM agent that models user goals, constraints, and preferences through interaction history and persistent memory, ToM-SWE improves performance on ambiguous and stateful SWE-bench benchmarks. Extensive evaluations and a three-week user study with professional developers demonstrate significant gains in task success rates and user satisfaction, highlighting the importance of stateful user modeling for effective human–AI collaboration in software engineering.

**Reviewer Concerns:**

Overall Concerns Summary: (1) Clarity & readability issues: Several figures (Figs. 1, 4–6) and metrics are hard to interpret due to unclear definitions, inconsistent terminology (resolution vs satisfaction), and missing explanations. The stateful SWE-bench setup and scoring need clearer, more concrete illustrations and examples. (2) Benchmark design & validity: The distinction between stateful SWE-bench and standard/ambiguous SWE-bench is insufficiently specified. Performance gains may largely stem from prompt underspecification and lack of interaction in baselines rather than ToM reasoning itself. (3) Baseline strength & ablations: The paper lacks strong baselines (e.g., single-agent memory, editable user profile, better prompting) to justify the necessity of a dual-agent ToM architecture. (4) User study limitations: Human study results are difficult to interpret without a control group (e.g., CodeAct). Acceptance and satisfaction metrics are not clearly tied to actual task resolution or final code quality. (5) Analysis gaps: Limited investigation of tradeoffs between user satisfaction and task correctness, correlation between metrics, and the actual causal impact of ToM interventions. (6)Reproducibility & transparency: Insufficient details on user simulator prompts, satisfaction scoring, developer profiles, and ToM–SWE interaction frequency hinder reproducibility. After the rebuttal, some concerns still exist.

**Reviewer Scores:**

All reviewers would keep their score unchaged

---

### Decision · Program_Chairs · 2026-01-26

Reject